# ALIGNMENT-AWARE DECODING

## ABSTRACT

Alignment of large language models remains a central challenge in natural language processing. Preference optimization has emerged as a popular and effective method for improving alignment, typically through training-time or prompt-based interventions. In this paper, we introduce alignment-aware decoding (AAD), a method to enhance model alignment directly at inference. Theoretically, AAD can be interpreted as implicit reward optimization, yet it requires no specialized training beyond the standard DPO setup. Empirically, AAD consistently outperforms strong baselines across diverse alignment benchmarks and model scales. Moreover, in data-constrained settings, AAD can produce high-quality synthetic data to improve alignment under standard decoding, providing a practical solution when labeled data is limited.

## 1 INTRODUCTION

Large language models (LLMs) are the backbone of modern natural language processing, powering applications ranging from open-ended dialogue to complex reasoning tasks. Despite their impressive capabilities, aligning these models with human preferences remains a central challenge. Misaligned models can produce harmful, biased, or simply unhelpful outputs, motivating a growing body of work on alignment, i.e., the process of training models to better reflect human values and preferences (Ziegler et al., 2019; Ouyang et al., 2022; Amodei et al., 2016).

Alignment is typically performed during training, either through reinforcement learning from human feedback (RLHF) or more recent variants such as direct preference optimization (DPO) (Rafailov et al., 2023). While these methods can achieve strong empirical results, they tend to be sensitive to imperfect preference signals. In RLHF, this arises from errors in the learned reward model that can be exploited (Amodei et al., 2016), while in DPO it stems from noise in the preference data itself (Rafailov et al., 2024a). To prevent over-optimization, the learned policy is typically constrained to remain close to a fixed reference model. This constrain ensures stability but also causes the optimal policy to inherit the biases of the reference model. This is because under this formulation, the learned policy is effectively trained as a reward model (Rafailov et al., 2023), and no longer as a policy that maximizes reward (Rafailov et al., 2024b).

An emerging alternative is *inference-time alignment*, which steers model outputs at inference, without modifying parameters. Recent work explores emulated fine-tuning (Mitchell et al., 2024; Liu et al., 2024a; Xu et al., 2025), energy-based decoding (Yuan et al., 2025; Hong et al., 2025), and value-guided search (Zhou et al., 2024; Liu et al., 2024e), all of which leverage reward signals to bias generation. These methods offer flexibility when model weights are frozen or proprietary, but often require auxiliary models, complex search procedures, or carefully tuned hyperparameters to remain stable.

In this paper, we introduce *Alignment-Aware Decoding* (AAD), a simple yet effective method to reliably improve alignment directly at inference. Our method leverages two distinct embedded features of the DPO-aligned model. First, its capacity to identify safe candidate tokens for the next decoding step via standard token likelihoods, and second, its ability to perform token-level credit assignment through the log-likelihood ratio with the reference model (Rafailov et al., 2024b). Intuitively, AAD exploits the alignment signal captured during preference optimization, which is often underutilized by standard decoding, and leverages the reference model at inference to mitigate biases it may have imparted to the aligned model, in a manner similar to methods that use a weaker (e.g., smaller) model to guide the decoding of a stronger model (Li et al., 2023a).

Empirically, we demonstrate that AAD consistently improves alignment across diverse benchmarks and model scales under compute-equivalent conditions. Furthermore, when high-quality preference data or inference resources are scarce, AAD can generate high-value synthetic completions that can be fed back into the model through iterative DPO (Pang et al., 2024), enabling stronger alignment without additional inference overhead.

We summarize our contributions as follows:

- We introduce alignment-aware decoding (AAD), a simple inference-time method that uses the aligned model as a token reward function. Importantly, AAD requires no additional training, using only the reference model (before DPO) and the aligned model (after DPO).
- We demonstrate across multiple benchmarks and model scales that AAD consistently and significantly improves alignment over baselines in compute-equivalent baselines.
- We further demonstrate that AAD can be used to generate high-quality synthetic data to further improve the alignment of LLMs under standard decoding strategies.

## 2 RELATED WORK

Recent efforts in aligning large language models (LLMs) with human preferences can be grouped into two broad categories: *training-time alignment* and *inference-time alignment*.

**Training-time alignment.** These approaches modify the model parameters to internalize the desired behavior directly during training. Reinforcement learning from human feedback (RLHF) is the standard paradigm for aligning LLMs (Ziegler et al., 2019), where a reward model is trained from human preferences and used to fine-tune the policy via a reinforcement learning algorithm such as proximal policy optimization (PPO) (Schulman et al., 2017). Direct preference optimization (DPO) (Rafailov et al., 2023) and variants (Hong et al., 2024; Azar et al., 2024; Ethayarajh et al., 2024; Zhao et al., 2023) eliminate the reinforcement learning stage of RLHF by optimizing a simple objective that compares preferred and dispreferred outputs. Building on this idea, selective DPO (Yang et al., 2024) improves sample efficiency by focusing the loss on key tokens with high preference signal. Weak-to-strong alignment (Zhu et al., 2025) further extends the paradigm by using a smaller, already aligned reference model to guide the training of a larger base model, thereby transferring alignment without costly reward modeling. From a theoretical perspective, Rafailov et al. (2024b) show that DPO training can be interpreted as learning a $Q$-function, which enables fine-grained credit assignment and provides a foundation for our method.

**Inference-time alignment.** Instead of modifying the base model, these methods steer generation on the fly, offering flexibility when model weights are frozen or inaccessible. One prominent line of work is *emulated fine-tuning* (EFT), where a reference–aligned model pair is used to define an implicit token-level reward function for decoding a third, unaligned base model (Mitchell et al., 2024), effectively emulating its alignment at inference. Such works include proxy alignment (Liu et al., 2024a), and GenARM (Xu et al., 2025), which differ mainly in how the token reward signal is estimated. Energy-based decoding (Yuan et al., 2025; Hong et al., 2025) takes a different angle by directly biasing generation toward low-energy regions of the reward and model's logit landscape. Loosely related, Liu et al. (2024d) introduce decoding-time realignment (DeRa), a decoding strategy that mimics DPO-trained models at different $\beta$ values without requiring retraining for each new value, and PAD (Chen et al., 2025), which integrates verbose preference signals into the reward. Closer to our work are methods that employ explicit rewards along with with lookahead search, such as DeAl (Huang et al., 2024), ARGS (Khanov et al., 2024), controlled decoding (Mudgal et al., 2023), and reward-guided beam search (Deng & Raffel, 2023). By contrast, our method does not rely on a separate explicit reward function. Tangential to our work are chunk-level value optimization methods that combine local search algorithm with external (implicit or explicit) value functions to select completions exhibiting the highest alignment. These include weak-to-strong decoding (Zhou et al., 2024), which generates candidate chunks with a base model and ranks them using an implicit value function derived from a reference–aligned model pair; IVG (Liu et al., 2024e), which generates chunks via EFT and ranks them with a learned value function and PPO-guided Monte Carlo tree search (Liu et al., 2024c), which reuses the value function obtained during PPO training to guide the search.

---

**Prompt:** What breed dog is smallest?

---

**Alignment-aware Decoding:** The smallest ● 1 recognized breed of dog is the Chihuahua. ● 2 Typically, ● 3 adult Chihuahuas ● 4 range ● 5 in size from 6 to 9 inches (15 to 23 centimeters) in height ● 6 at the shoulder and ● 7 can weigh ● 8 between 2.5 to 6 pounds ● 9 (1.13 to 2.72 ● 10 kg). ● 11 They are known for their ● 12 tiny size and ● 13 distinctive appearance, with ● 14 a ● 15 sturdy build and ● 16 large ● 17 heads relative to their ● 18 body size.

---

**Greedy Continuations:**
● 1 breed of dog is the Chihuahua. They typically weigh between 2 to 6 pounds and stand...
● 2 They typically weigh between 2 to 6 pounds and stand 6 to 9 inches tall at the...
● 3 they weigh between 2 to 6 pounds and stand 6 to 9 inches tall at the shoulder.
● 4 weigh between 2 and 6 pounds (0.9 to 2.7 kg), and they can be as small as 3 inches (
● 5 from 6 to 9 inches (15 to 23 cm) in height and weigh between 2.5 and 6 pounds (1.
● 6 and weigh between 2.5 and 6 pounds (1.1 to 2.7 kilograms). However, there are some...
● 7 weigh between 2.5 and 6 pounds (1.1 to 2.7 kilograms). However, there are some very...
● 8 anywhere from 2 to 6 pounds (0.9 to 2.7 kilograms).
● 9 (1.1 to 2.7 kilograms). However, there are some very small Chihuahuas that can be...
● 10 kilograms). However, there are some very small Chihuahuas that can be even smaller...
● 11 However, there are some very small Chihuahuas that can be even smaller than this.
● 12 small size and big personalities.
● 13 big personalities.
● 14 their long snouts and large, round eyes.
● 15 long snout, large ears, and a distinctive coat.
● 16 a long, pointed snout.
● 17 , expressive eyes.
● 18 bodies.

---

**Best-of-2:** The smallest breed of dog is the Chihuahua. The smallest Chihuahua, a dog that weighed less than 1 pound, named MiMi, held the title of the world's smallest dog from October 2, 2010, to November 21, 2012. Since then, she is considered to be the smallest dog in terms of weight. Some Chihuahuas can weigh up to 6 pounds or more, but MiMi's small size made her a unique and famous dog.

Figure 1: **Qualitative comparison of AAD against other decoding strategies**. Greedy continuations are generated by feeding the prompt together with the current AAD prefix back into the model and greedily selecting the next token, revealing where the greedy trajectory diverges from AAD. For instance, at ● 8, given a context up to [...] can weigh, AAD generates between 2.5 to 6 pounds [...] while greedy generates anywhere from 2 to 6 pounds [...]. AAD identifies the Chihuahua as the smallest recognized breed of dog, making the distinction that it refers to an officially recognized classification, whereas the greedy continuation and Bo2 simply state breed without that nuance. AAD is also the only method that directly addresses size (the core of the prompt) by describing height and body proportions, while greedy and Bo2 focus mainly on weight. This highlights AAD's advantage in preserving prompt adherence.

## 3 BACKGROUND

**Auto-regressive language modeling.** Let $\mathcal{V}$ denote the token vocabulary, and let $\pi$ denote an auto-regressive language model (LM) which, given a context $x$, generates a sequence $y$ with probability $\pi(y \mid x) = \prod_{t=1}^{|y|} \pi(y_t \mid x \circ y_{1:t-1})$, where $y_{1:t}$ denotes the prefix of $y$ up to and including position $t$, and $\circ$ denotes sequence concatenation ($y_{1:0} = \emptyset$ by convention). Training $\pi$ typically involves three phases (Ziegler et al., 2019; Ouyang et al., 2022): (i) *pretraining*, (ii) *supervised fine-tuning* (SFT), and (iii) *preference optimization* (PO). During pretraining, the model is trained on large-scale unlabeled corpora to predict the next token given a prefix of text. Then, this model is generally fine-tuned on curated, task-specific datasets through supervised learning, which improves its ability to follow instructions and generate useful outputs in more constrained settings (e.g., chatbot dialogue, summarization). For the remainder of this work, we denote by $\pi_{\text{SFT}}$ the model obtained after SFT. While such models can follow instructions, they often produce outputs that are suboptimal with respect to human values and preferences. PO further adapts $\pi_{\text{SFT}}$ to better reflect these preferences.

**Preference optimization.** The goal of PO is to align the model with a conditional preference relation $\succ_x$, with $y_1 \succ_x y_2$ indicating that the completion $y_1$ is preferred over $y_2$ given the prompt $x$. In practice, preference relations are typically modeled probabilistically using the Bradley-Terry (BT) model (Bradley & Terry, 1952), which posits the existence of a scoring function $r^*$ that quantifies

Table 1: **Performance of AAD** across datasets, with decoding methods as rows and base models as columns. Each cell reports the average oracle reward ($R$) and AAD's win rate ($W$) against the corresponding method. Higher values indicate better alignment. AAD consistently achieves the highest rewards and win rate across all settings, demonstrating its strong alignment capability.

| Method | Models & Datasets | | | | | | | |
|---|---|---|---|---|---|---|---|---|
| | Llama 3B | | Llama 8B | | Qwen 0.6B | | Qwen 4B | |
| | $R$ | $W$ [%] | $R$ | $W$ [%] | $R$ | $W$ [%] | $R$ | $W$ [%] |
| | *Ultrachat* | | | | | | | |
| Greedy SFT | 0.58 | 0.86 | 0.87 | 0.85 | -0.88 | 0.80 | 0.22 | 0.80 |
| Greedy DPO | 0.68 | 0.86 | 0.98 | 0.84 | -0.69 | 0.78 | 0.29 | 0.79 |
| Bo2 | 0.85 | 0.85 | 1.06 | 0.85 | -0.62 | 0.78 | 0.47 | 0.77 |
| EFT | 1.04 | 0.83 | 1.27 | 0.81 | -0.19 | 0.67 | 0.58 | 0.73 |
| AAD (ours) | **2.21** | - | **2.22** | - | **0.34** | - | **1.19** | - |
| | *Agrilla* | | | | | | | |
| Greedy SFT | 1.59 | 0.88 | 1.72 | 0.89 | -0.86 | 0.89 | 0.70 | 0.87 |
| Greedy DPO | 2.48 | 0.86 | 2.55 | 0.87 | 0.12 | 0.80 | 1.37 | 0.82 |
| Bo2 | 3.02 | 0.84 | 3.16 | 0.86 | 0.68 | 0.77 | 1.94 | 0.78 |
| EFT | 4.54 | 0.70 | 4.65 | 0.72 | 1.99 | 0.52 | 3.28 | 0.61 |
| AAD (ours) | **5.64** | - | **5.90** | - | **2.33** | - | **3.84** | - |
| | *OpenRLHF Mixture* | | | | | | | |
| Greedy SFT | 3.59 | 0.90 | 3.89 | 0.93 | 0.83 | 0.83 | 2.63 | 0.88 |
| Greedy DPO | 4.54 | 0.88 | 4.93 | 0.89 | 1.74 | 0.76 | 3.56 | 0.79 |
| Bo2 | 5.34 | 0.83 | 5.60 | 0.85 | 2.42 | 0.68 | 4.48 | 0.69 |
| EFT | 6.18 | 0.72 | 6.84 | 0.67 | 3.08 | 0.55 | 5.29 | 0.54 |
| AAD (ours) | **7.28** | - | **7.60** | - | **3.42** | - | **5.45** | - |
| | *HHRLHF* | | | | | | | |
| Greedy SFT | -1.89 | 0.62 | -1.13 | 0.61 | -1.36 | 0.65 | -0.53 | 0.64 |
| Greedy DPO | -1.83 | 0.61 | -1.08 | 0.60 | -1.25 | 0.60 | -0.49 | 0.63 |
| Bo2 | -1.65 | 0.64 | -0.91 | 0.61 | -1.06 | 0.64 | -0.22 | 0.59 |
| EFT | -1.74 | 0.61 | -0.98 | 0.57 | -1.12 | 0.57 | -0.47 | 0.64 |
| AAD (ours) | **-0.97** | - | **-0.34** | - | **-0.61** | - | **-0.02** | - |
| | *Skywork* | | | | | | | |
| Greedy SFT | 7.93 | 0.74 | 13.25 | 0.80 | -4.41 | 0.66 | 9.34 | 0.75 |
| Greedy DPO | 8.45 | 0.72 | 13.64 | 0.78 | -3.73 | 0.66 | 9.54 | 0.74 |
| Bo2 | 9.04 | 0.74 | 14.15 | 0.76 | -5.18 | 0.73 | 9.35 | 0.76 |
| EFT | 10.03 | 0.68 | 15.57 | 0.72 | -1.88 | 0.58 | 10.35 | 0.71 |
| AAD (ours) | **13.71** | - | **19.27** | - | **-0.01** | - | **14.44** | - |
| | *Nectar* | | | | | | | |
| Greedy SFT | 0.72 | 0.99 | 1.17 | 0.99 | -0.77 | 0.93 | 0.77 | 0.94 |
| Greedy DPO | 1.45 | 0.98 | 2.12 | 0.99 | 0.09 | 0.84 | 1.45 | 0.85 |
| Bo2 | 2.15 | 0.95 | 2.64 | 0.93 | 1.07 | 0.70 | 1.99 | 0.74 |
| EFT | 2.28 | 0.89 | 3.30 | 0.75 | 1.23 | 0.58 | 2.35 | 0.65 |
| AAD (ours) | **3.63** | - | **3.70** | - | **1.68** | - | **2.71** | - |

the quality of a prompt-completion pair $(x, y)$. Specifically, with $\sigma(z) = (1 + e^{-z})^{-1}$ denoting the sigmoid function, the BT model defines the likelihood of $y_1$ being preferred over $y_2$ given $x$ as

$$p(y_1 \succ_x y_2) = \sigma\big(r^*(x, y_1) - r^*(x, y_2)\big), \tag{1}$$

and therefore provides a likelihood-based framework to train the LM on observed preferences. Starting from $\pi_{\text{SFT}}$ and a prompt distribution $\rho$, the training objective of PO can be formulated as the KL-constrained optimization problem (Jaques et al., 2017):

$$\pi^* = \arg\max_{\pi} \mathbb{E}_{x \sim \rho} \big[\mathbb{E}_{y \sim \pi(\cdot|x)} [r^*(x, y)] - \beta D_{\text{KL}}\big(\pi(\cdot \mid x) \| \pi_{\text{SFT}}(\cdot \mid x)\big)\big], \tag{2}$$

with $\beta > 0$ a hyperparameter preventing overoptimization. The classical approach to solving Eq. (2) is known as reinforcement learning from human feedback (RLHF), and proceeds in two steps (Ziegler et al., 2019). First, a parametric reward model $r_\theta(x, y)$ is trained to minimize the negative log likelihood of observed preferences:

$$\mathcal{L}(r_\theta; \mathcal{D}) = -\frac{1}{|\mathcal{D}|} \sum_{i=1}^{|\mathcal{D}|} \log \sigma\big(r_\theta(x^i, y_w^i) - r_\theta(x^i, y_l^i)\big), \tag{3}$$

where $\mathcal{D} = \{(x^i, y_w^i, y_l^i) \mid x^i \sim \rho, \; y_w^i \succ_{x^i} y_l^i\}$ is a static preference dataset. In a second stage, Eq. (2) is approximately solved using policy gradient methods, such as PPO (Schulman et al., 2017), on a parametric class of models. Despite their effectiveness, reinforcement learning algorithms are prone to reward hacking (Amodei et al., 2016) and typically require generating many rollouts during training, which can be computationally expensive and unstable. To address these challenges, Rafailov et al. (2023) introduce *direct preference optimization* (DPO) to directly approximate $\pi^*$ via a supervised objective. Formally, they note that the closed form solution of Eq. (2) can be expressed in terms of the optimal policy as

$$\pi^*(y \mid x) = \frac{1}{Z(x)} \pi_{\text{SFT}}(y \mid x) \exp\left(\frac{1}{\beta} r^*(x, y)\right), \tag{4}$$

with $Z(x; r^*) = \sum_{y'} \pi_{\text{SFT}}(y' \mid x) \exp(\frac{1}{\beta} r^*(x, y'))$ the partition function. Rearranging the terms, they find that the $r^*$ must satisfy

$$r^*(x, y) = \beta \log \frac{\pi^*(y \mid x)}{\log \pi_{\text{SFT}}(y \mid x)} + \beta \log Z(x; r^*). \tag{5}$$

The key idea of DPO is to eliminate the second stage of RLHF by directly minimizing Eq. (3) within a restricted reward class $r_\theta(x, y) = \beta \log \frac{\pi_\theta(y|x)}{\pi_{\text{SFT}}(y|x)}$. This choice ensures that $Z(x, r_\theta) = 1$, such that, as per Eq. (5), $r_\theta(x, y) = r^*(x, y)$ if and only if $\pi_\theta(y \mid x) = \pi^*(y \mid x)$. In other words, the model obtained after DPO, $\pi_{\text{DPO}} \approx \pi^*$, is simply a byproduct from training the reward model $r_\theta$ on the preference dataset $\mathcal{D}$.

## 4 METHOD

**The aligned policy $\pi^*$ inherits the biases of $\pi_{\text{SFT}}$.** The main motivation behind PO is that it increases the likelihood of completions with higher rewards, as shown in Eq. (4). However, counter-intuitively, even the optimal analytical solution $\pi^*$ can sometimes favor a completion with a lower reward over one with a higher reward. To illustrate this, let $x$ be a prompt and $y_1, y_2$ any two completions satisfying $r^*(x, y_1) \geq r^*(x, y_2)$. From Eq. (4), we have

$$\log \frac{\pi^*(y_1 \mid x)}{\pi^*(y_2 \mid x)} = \underbrace{\log \frac{\pi_{\text{SFT}}(y_1 \mid x)}{\pi_{\text{SFT}}(y_2 \mid x)}}_{:=\Delta_{\text{SFT}}} + \frac{1}{\beta} \underbrace{\left(r^*(x, y_1) - r^*(x, y_2)\right)}_{:=\Delta_r}. \tag{6}$$

This implies that if $\Delta_{\text{SFT}} < -\frac{1}{\beta} \Delta_r$, then $\pi^*(y_1 \mid x) \leq \pi^*(y_2 \mid x)$ although $y_1$ is preferred over $y_2$ given $x$. In other words, the optimal model $\pi^*$ inherits the biases of $\pi_{\text{SFT}}$. Note that this is not due to reward hacking as we only consider the exact reward $r^*$ is our derivation. This is consistent with the observation of Rafailov et al. (2024b) that PO does not train a policy to directly maximize reward.

**Token-level reward.** We propose to use $\pi_{\text{DPO}}$ together with $\pi_{\text{SFT}}$ as an approximate token-level advantage function. Generating a completion $y$ given a context $x$ then amounts to maximizing

$$r_{\text{DPO}}(x, y) = \beta \frac{\pi_{\text{DPO}}(y \mid x)}{\pi_{\text{SFT}}(y \mid x)}.$$

Since exact maximization is intractable, we use greedy decoding with a token-level advantage function

$$A(v \mid x \circ y_{1:t}) = \log \frac{\pi_{\text{DPO}}(v \mid x \circ y_{1:t})}{\pi_{\text{SFT}}(v \mid x \circ y_{1:t})} \qquad v \in \mathcal{V}. \tag{7}$$

Decoding according to this ratio provides a more direct and human-aligned way to maximize reward than following $\pi_{\text{DPO}}$ directly. A detailed theoretical justification for this behavior can be found in Appendix A.1. We omit $\beta$ as it does not change the ranking of candidate sequences.

**Preventing over-optimization.** Since DPO is trained on relatively small datasets, the advantage $A$ may be unreliable for low-probability tokens, and maximizing it without constraints at each decoding step typically produces degenerate completions. For instance, tokens that are essential for grammatical and semantic coherence might be assigned high probabilities by both $\pi_{\text{DPO}}$ and $\pi_{\text{SFT}}$,

---

**Algorithm 1:** Alignment-Aware Decoding (AAD)

---

**Input:** DPO model $\pi_{\text{DPO}}$ with its SFT version $\pi_{\text{SFT}}$, prompt $x$, max length $T$, threshold $\alpha$

**for** $t = 1$ **to** $T$ **do**

$\quad$ Compute $\pi_{\text{DPO}}(\cdot \mid x \circ y_{1:t-1})$ and $\pi_{\text{SFT}}(\cdot \mid x \circ y_{1:t-1})$ ; $\quad$ // Model distributions

$\quad$ Compute $A(\cdot \mid x \circ y_{1:t-1})$ from Eq. (7) ; $\qquad\qquad$ // Reward-based score

$\quad$ $p_{max} \leftarrow \max_{v \in \mathcal{V}} \pi_{\text{DPO}}(v \mid x \circ y_{1:t-1})$ ; $\qquad\qquad$ // Max probability

$\quad$ $\mathcal{V}_\alpha \leftarrow \{ v \in \mathcal{V} \mid \pi_{\text{DPO}}(v \mid x \circ y_{1:t-1}) \geq \alpha p_{max} \}$ ; $\qquad$ // Token filter

$\quad$ $y_t \leftarrow \arg\max_{v \in \mathcal{V}_\alpha} A(v \mid x \circ y_{1:t-1})$ ; $\qquad\quad$ // Alignment-aware choice

$\quad$ **if** $y_t = \texttt{<eos>}$ **then**

$\quad\quad$ **return** $y_{1:t-1}$ ; $\qquad\qquad\qquad\qquad$ // Stop if EOS is generated

**return** $y_{1:T}$ ; $\qquad\qquad\qquad\qquad\qquad$ // Return truncated sequence

---

making their ratio too small to be selected under the proposed decoding algorithm. Moreover, if $\pi_{\text{sft}}$ assigns a small probability to a given token, even a tiny absolute increase from PO training can produce a large relative change, leading to spuriously high scores and numerical instabilities. To mitigate these issues, we take inspiration from contrastive decoding (Li et al., 2023a), a decoding algorithm that uses a small model to boost the performance of a larger one, and apply min-$\alpha$ filtering to the DPO probabilities $\pi_{\text{DPO}}$ (Minh et al., 2025), restricting the alignment-aware decoding to plausible tokens only.

**Proposed method: alignment-aware decoding (AAD).** Formally, alignment-aware decoding selects the token at position $t$ according to

$$y_t = \underset{v \in \mathcal{V}_\alpha(x \circ y_{1:t-1})}{\arg\max} A(v | x \circ y_{1:t-1}), \tag{8}$$

where

$$\mathcal{V}_\alpha(x \circ y_{1:t-1}) = \{ v \in \mathcal{V} \mid \pi_{\text{DPO}}(v | x \circ y_{1:t-1}) \geq \alpha \max_{v' \in \mathcal{V}} \pi_{\text{DPO}}(v' | x \circ y_{1:t-1}) \} \subseteq \mathcal{V} \tag{9}$$

is the set of plausible token over which alignment can safely be optimized. A pseudocode for AAD is presented in Algorithm 1.

## 5 EXPERIMENTAL SETUP

**Overview.** We conduct a series of experiments to evaluate the effectiveness of our method against several baselines. Each experiment begins with a preference dataset, which serves as the foundation for training both reward and aligned models. We split the data into a 90/10 training/evaluation set. An oracle reward model is trained on the full training split. In parallel, we subsample 10% of the training split for two purposes: (i) training a picker reward model and (ii) aligning a SFT model $\pi_{\text{SFT}}$ via DPO to obtain $\pi_{\text{DPO}}$. This setup allows to simulate two conditions simultaneously: the availability of a strong oracle reward model for evaluation, and the scarcity of preference data, which is typically costly and difficult to obtain. The picker reward model is then used to select the highest-scoring continuation in methods such as best-of-$N$ (Bo$N$) sampling. For evaluation, we sample a fixed number of prompts from the validation split and generate continuations using both our method and the baselines. These continuations are scored with the oracle reward model. Evaluation metrics include (i) the win rate ($W$) of our method over a baseline, computed via pairwise continuation comparisons, and (ii) the average oracle reward ($R$) across all generated outputs. In addition, we also evaluate our method using the external AlpacaEval framework (Li et al., 2023b). For reproducibility, we refer the reader to Appendix A.6, which contains the link to our codebase.

**Datasets and reward models.** For training and evaluation, we use preference datasets that are commonly adopted in reward modeling, including Ultrachat (Ultrachat, 2025), Argilla (Cui et al., 2023), the OpenRLHF Mixture (Dong et al., 2023; Xiong et al., 2024), HHRLHF (Bai et al., 2022), Nectar (Zhu et al., 2023), and Skywork (Liu et al., 2024b). For the first 4 datasets, we train the reward models (pickers and oracles) using the training procedure detailed below. For Skywork and Nectar, we do not train the oracles and instead follow a specialized evaluation protocol: prompts

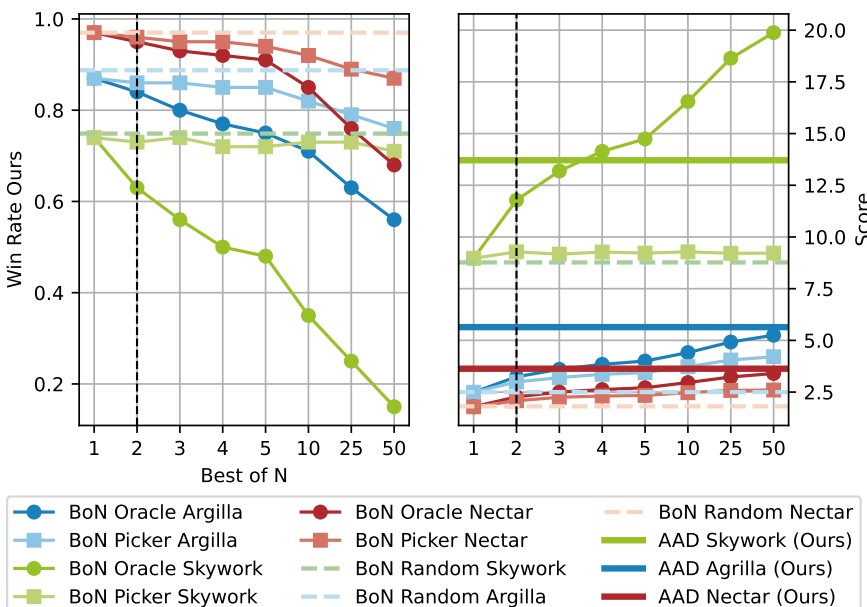

Figure 2: **AAD versus Bo$N$**. We evaluate AAD against three selection strategies on Argilla, Nectar and Skywork datasets for different values of $N$: (i) Bo$N$ using the oracle, (ii) Bo$N$ using the picker, and (iii) random selection among $N$ completions. AAD remains competitive even against Bo$N$-Oracle reward model, a setting that is by design unfavorable to AAD, since the oracle is used both for Bo$N$ selection and evaluation, whereas AAD only uses a model aligned on 10% of the data. On Skywork, Bo$N$ reaches the performance of AAD for $N = 4$ but requires roughly twice as much compute. On Argilla and Nectar, even $N = 50$ fails to match AAD's performance. The vertical dashed line indicates the point at which the computational cost of Bo$N$ matches that of our method. For the random selection baseline, we report only the mean performance across all test runs.

are drawn from the AlpacaEval dataset (Li et al., 2023b), and scores are assigned using off-the-shelf oracle reward models trained externally on the respective datasets. Specifically, for Skywork we use the Skywork reward model, which is based on Llama-3.1-8B, and for Nectar we use the Starling reward model, which is based on Llama2-7B-Chat. This ensures that the oracle has not been trained on the prompts used for evaluation. At the time of writing, the oracle reward model for Skywork is the best-performing reward model in Reward Bench (Malik et al., 2025), a standardized framework for evaluating reward models.

**Training.** We train both the pickers (for all datasets) and oracle reward models (except for Skywork and Nectar) using full fine-tuning with an additional classification layer, optimized under the Bradley-Terry loss detailed in Eq. (3). Training is performed for two epochs. For the aligned models $\pi_{\text{DPO}}$, we also conduct two epochs of training, employing LoRA adapters (Hu et al., 2021) to ensure parameter efficiency and regularization. Comprehensive training details are provided in Appendix A.3. The accuracies of the oracle and picker reward models on the evaluation splits of the datasets are reported in Appendix A.2.

**Baselines.** For evaluation, we compare our method against four alternative decoding strategies that only use $\pi_{\text{DPO}}$, $\pi_{\text{SFT}}$, or both: (i) greedy decoding with $\pi_{\text{SFT}}$, (ii) greedy decoding with $\pi_{\text{DPO}}$, (iii) Bo2 sampling with $\pi_{\text{DPO}}$, and (iv) a variation of EFT (Mitchell et al., 2024; Liu et al., 2024a; Rafailov et al., 2024b) using $\pi_{\text{SFT}}$ for both the base and reference model, and setting $\beta = 4$, which has been found to perform the best across multiple settings. For (iii), two candidate responses are generated with the aligned model via nucleus (top-$p$) sampling with $p = 0.9$ (Holtzman et al., 2020), after which the picker reward model of the corresponding preference dataset selects the higher-scoring output. Both (iii) and (iv) entail a computational cost comparable to our method, whereas (i) and (ii) incur roughly half that cost.

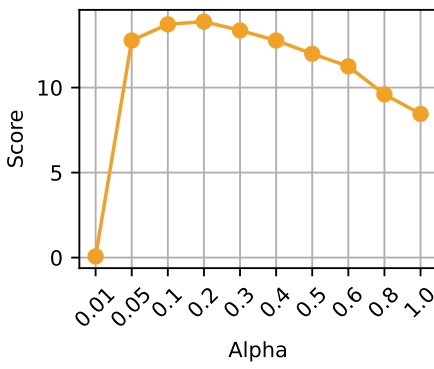 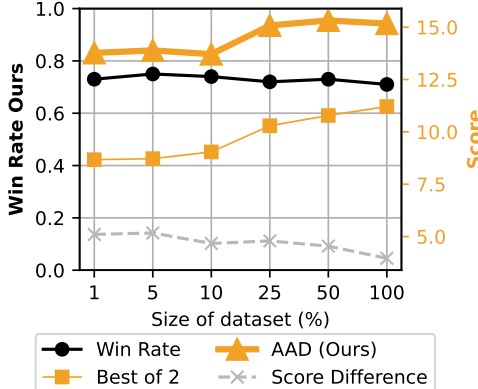

Figure 3: **Effect of the hyperparameter $\alpha$ in AAD**. The results were obtained using a 3B LLaMA model trained on the Skywork dataset with LoRA finetuning. Model performance exhibits a clear peak for $\alpha$ values in the range of approximately 0.1 to 0.2.

Figure 4: **Performance of AAD across different training dataset sizes** on the Skywork dataset. Results show that AAD consistently outperforms Bo2 at every data scale, providing clear evidence of its robustness in low-data regimes.

**Generation.** For decoding with the AAD method, we set the token filtering parameter $\alpha = 0.1$ as defined in Eq. (9) and ablated in Fig. 3. Across all decoding methods, the `<user>` token is treated as an end-of-sequence marker.

## 6 RESULTS

**AAD consistently outperforms baselines.** The main results of our experiments are presented in Table 1. Across both model families, our method consistently outperforms the baselines by a substantial margin, achieving notably strong win rates with larger models. Remarkably, our method continues to deliver strong gains even when evaluated with external oracle reward models (Nectar and Skywork). On the AlpacaEval framework (see Table 2), our method also achieves mostly high win rates. We also test AAD against traditional contrastive decoding (Li et al., 2023a), where we use an amateur model instead of the SFT version in Eq. (6). The results in Table 3 strongly suggest that the SFT model is crucial for the alignment improvements achieved by AAD. We provide additional results for AAD in Appendix A.4, including comparisons to beam search, the effect of training epochs, and results with fully fine-tuned models.

**Correspondence between BoN and AAD.** In BoN sampling, the expected reward of the selected sequence increases as $N$ grows, since sampling more candidates raises the likelihood of obtaining a higher-scoring response by chance. Fig. 2 shows that our method remains competitive even when compared against BoN sampling with the oracle reward model, despite the oracle being trained on ten times more data than $\pi_{\text{DPO}}$, and despite the oracle being also used for the evaluation.

Table 2: **AAD win rate on AlpacaEval** with default evaluator (GPT-4) across Skywork and Nectar (Li et al., 2023b). AAD consistently matches or outperforms baselines.

| Method | Skywork | | | | Nectar | | | |
|---|---|---|---|---|---|---|---|---|
| | Llama 3B | Llama 8B | Qwen 0.6B | Qwen 4B | Llama 3B | Llama 8B | Qwen 0.6B | Qwen 4B |
| Greedy SFT | 0.77 | 0.79 | 0.74 | 0.76 | 0.80 | 0.82 | 0.52 | 0.61 |
| Greedy DPO | 0.76 | 0.77 | 0.73 | 0.75 | 0.76 | 0.76 | 0.44 | 0.54 |
| Bo2 | 0.75 | 0.78 | 0.73 | 0.77 | 0.76 | 0.72 | 0.48 | 0.50 |
| EFT | 0.73 | 0.73 | 0.65 | 0.73 | 0.70 | 0.63 | 0.44 | 0.50 |

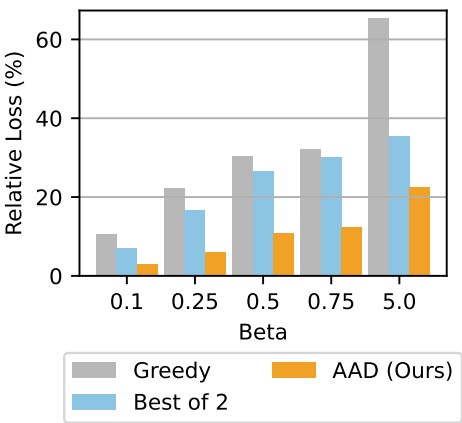

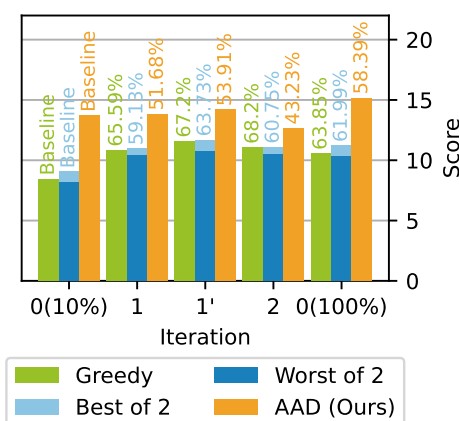

Figure 5: **Relative alignment loss** of the oracle score $R$ on the Agrilla dataset as a function of the DPO regularization parameter $\beta$, with baseline performance established at $\beta = 0.05$. As expected, across all strategies, larger $\beta$ values reduce alignment, but AAD consistently shows the lowest relative loss, demonstrating greater hyperparameter robustness compared to baselines. This behavior stems from the fact that $r^*$ is $\beta$-independent, but $\pi^*$ is not, as seen in Section 3.

Figure 6: **Effect of iterative DPO**. Results show that iterative DPO using AAD-generated data substantially improves alignment, approaching full-dataset performance (100%) with only 10% of the original data. Win rates against the original $\pi_{\mathrm{DPO}}$ (baseline), using the same decoding scheme, are shown above the bars. Iteration $i$ indicates the average oracle score of a model that has undergone DPO using AAD-generated data, initiated from $\pi_{\mathrm{SFT}}$ for $i = 1, 2$ or $\pi_{\mathrm{DPO}}$ for $i = 1'$.

**Stabilizing beam search via entropy thresholding.** We also investigate if we can use beam search on the token reward defined in Eq. (7), rather than simply greedy maximization. Beam search typically suffers from beam collapse, and increasing the number of beams does not always improve generations, a phenomenon reminiscent of inference-time overoptimization. However, we find that (i) increasing $\alpha$ and (ii) introducing an entropy threshold can make beam search beneficial in some cases. The key observation is that certain tokens are highly predictable and thus are assigned high probability by both $\pi_{\mathrm{SFT}}$ and $\pi_{\mathrm{DPO}}$. In such cases, applying our score difference may incorrectly override an obvious continuation. To prevent this, we only apply our scoring adjustment when the aligned model is uncertain, that is, when the predictive entropy exceeds the threshold $\tau$. In practice, this is equivalent to setting $\pi_{\mathrm{SFT}}(y' \mid x) = 1/|\mathcal{V}_\alpha(x)|$ for every token $y' \in \mathcal{V}_\alpha(x)$ when the entropy $\sum_{y' \in \mathcal{V}_\alpha(x)} -\pi_{\mathrm{SFT}}(y' \mid x) \log \pi_{\mathrm{SFT}}(y' \mid x) \leq \tau$. The results in Fig. 7, obtained on the Skywork dataset with $\alpha = 0.7$, show that without entropy thresholding, scores rapidly degrade as the number of beams increases. By contrast, introducing the threshold stabilizes performance and makes beam search beneficial.

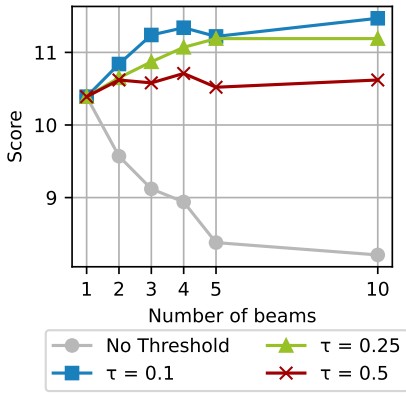

Figure 7: **Effect of beam size and entropy threshold** on performance for the Skywork dataset with $\alpha = 0.7$. Without entropy thresholding, scores rapidly degrade as the number of beams increases due to beam collapse. This mechanism enables larger beam sizes to yield improved alignment, while also reducing the computational cost compared to standard beam search.

**AAD performs strongly under data scarcity.** To assess our method in different data regimes, we train a series of picker reward models and aligned models on the Skywork dataset, gradually increasing the training data up to the full training split. We then evaluate our method against Bo2 sampling using the oracle. Results are shown in Fig. 4. Note that the 100 % mark in our plots does not represent the entire dataset used to train the external Skywork reward model, as we only

Table 3: **Comparison between AAD and contrastive decoding (CD)** (Li et al., 2023a) on weaker reference models across the Argilla, Skywork, and Nectar datasets. All methods decode with LLaMA-8B DPO as the main model. CD subtracts logits from external LLaMA Instruct models (1B and 3B), treating them as generic "amateur" references, while AAD uses the 8B SFT model that the DPO model was trained from. Reported metrics include the reward-model score ($R$) and win rate ($W$) of AAD. AAD consistently achieves the highest reward-model scores across all datasets.

| Method | Argilla | | Skywork | | Nectar | |
|---|---|---|---|---|---|---|
| | $R$ | $W$ [%] | $R$ | $W$ [%] | $R$ | $W$ [%] |
| CD 1B Instruct | 2.79 | 0.85 | 13.08 | 0.81 | 2.65 | 0.90 |
| CD 3B Instruct | 2.46 | 0.86 | 10.88 | 0.88 | 2.51 | 0.93 |
| AAD (ours) | **5.9** | - | **19.27** | - | **3.7** | - |

trained on the 90% training split and kept 10% for evaluation. Interestingly, AAD's win rate remains relatively consistent, suggesting that its performance generalizes across different data regimes.

**Effect of DPO regularization parameter $\beta$.** The $\beta$ parameter constitutes a critical regularization hyperparameter in DPO training. To assess its influence on our method, we establish baseline performance at $\beta = 0.05$ and evaluate the relative loss of models trained with $\beta$ values of 0.1, 0.25, 0.5, 0.75, and 5.0. We conduct these experiments on the Agrilla dataset. The evaluation is conducted under three decoding strategies: Bo2 sampling, greedy decoding and our method. The corresponding results are presented in Fig. 5. Across all strategies, larger $\beta$ values are associated with reduced alignment performance. Nevertheless, our decoding method consistently exhibits the lowest relative loss, indicating greater robustness and stability compared to the alternative approaches.

**Overcoming data scarcity with iterative DPO.** Since AAD appears to generate data with high alignment, we investigate if this data can be used to further train the aligned model. To this end, we implement a version of iterative DPO (Pang et al., 2024). We begin with our model $\pi_{\text{DPO}}$, trained solely on 10% of the original preference dataset (0th iteration), and using LLaMA3.2-3B-SFT (Lacoste et al., 2019) for $\pi_{\text{SFT}}$. In the first iteration, we construct a synthetic preference dataset using the prompts of the subsampled dataset, and by pairing completions as follows: chosen samples are generated with AAD, while rejected samples are produced via nucleus sampling on $\pi_{\text{DPO}}$ with hyperparameter 0.9. We then retrain DPO alignment on this synthetic dataset in two variants: (i) starting from the base model LLaMA3.2-3B-SFT (1st iteration) and (ii) starting from the model already aligned on the 10% preference dataset (1' iteration). We further extend this process with a second iteration. Here, we retain the rejected samples from the previous step and generate new chosen samples using our method in combination with the DPO model trained from LLaMA3.2-3B-SFT during the 1st iteration. This produces a new synthetic dataset, which is again used to retrain DPO alignment from the base LLaMA3.2-3B-SFT model (2nd iteration). Results shown in Fig. 6 highlight the significant benefits of iterative DPO. Remarkably, even with only 10% of the preference data, this method nearly closes the gap with a model trained on the full dataset.

# 7 CONCLUSION

We introduce *alignment-aware decoding* (AAD), a decoding strategy that treats a DPO-trained model as a token-level reward function. AAD performs on-the-fly implicit reward optimization without additional training or external models. Across multiple datasets and model families, we show that AAD consistently improves alignment while maintaining efficiency comparable to standard decoding. AAD can also generate high-quality synthetic aligned data, enabling iterative preference optimization under data scarcity. While AAD improves alignment, there are limitations; it requires two forward passes per token, as well as access to the original SFT model. Future directions include combining AAD with more sophisticated search strategies, exploring adaptive token filtering and entropy-based thresholds, and extending to other modalities such as image generation. Overall, we hope this work motivates further research on inference-time alignment methods that are both theoretically grounded and practically deployable.

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

# A APPENDIX

## A.1 DETAILED THEORETICAL JUSTIFICATION OF AAD

Our theoretical justification is based on the approach of Rafailov et al. (2024b). First, we model sequence generation as a token-level Markov Decision Process (MDP) $\mathcal{M} = (\mathcal{S}, \mathcal{V}, f, r_H, \rho)$, where $\mathcal{S}$ is the set of partial sequences $s_t = x \circ y_{1:t}$, $\mathcal{V}$ is the vocabulary of tokens, and the dynamics $f(s, a) = s \circ a$ deterministically append the selected token (action) $a$ to the current sequence $s$. The initial state distribution $\rho$ corresponds to prompts $x$, and the reward $r(s_t, a_t)$ defines the optimization problem. By convention, it is zero for all action if $s_t$ contains an end-of-sequence token <eos>. Typically, $r_H$ is such that $\sum_{t=0}^{T} r_H(s_t, a_t) = r^*(s_0, a_{0:T})$ reflects some sort of human preference for the complete sequence according to Eq. (1), with $T$ the index of the first <eos> token. Within this framework, one can rewrite the PO objective in Eq. (2) as

$$\pi^* = \arg\max_{\pi} \mathbb{E}_{x \sim \rho} \left[ \mathbb{E}_{y \sim \pi(\cdot|x)} [r^*(x, y)] - \beta D_{\mathrm{KL}} \big( \pi(\cdot \mid x) \| \pi_{\mathrm{SFT}}(\cdot \mid x) \big) \right] \tag{10}$$

$$= \arg\max_{\pi} \mathbb{E}_{x \sim \rho,\, y \sim \pi(\cdot|x)} \left[ r^*(x, y) - \beta \big( \log \pi(y \mid x) - \log \pi_{\mathrm{SFT}}(y \mid x) \big) \right] \tag{11}$$

$$= \arg\max_{\pi} \mathbb{E}_{s_0 \sim \rho,\, \tau \sim \pi(\cdot|s_0)} \left[ \sum_{t=0}^{T} (r_H(s_t, a_t) + \beta \log \pi_{\mathrm{SFT}}(a_t \mid s_t)) \right] - \beta \mathbb{E}_{x \sim \rho} \mathcal{H} \big( \pi(\cdot \mid x) \big) \tag{12}$$

$$= \arg\max_{\pi} \mathbb{E}_{s_0 \sim \rho,\, \tau \sim \pi(\cdot|s_0)} \left[ \sum_{t=0}^{T} r_{\mathrm{KL}}(s_t, a_t) \right] - \beta \mathbb{E}_{x \sim \rho} \mathcal{H} \big( \pi(\cdot \mid x) \big), \tag{13}$$

where $r_{\mathrm{KL}}(s_t, a_t) = r_H(s_t, a_t) + \beta \log \pi_{\mathrm{SFT}}(a_t \mid s_t)$ is the effective reward of the constrained MDP. We use the notation $\tau, s, a$ and $x, y$ for token-/sequence-level quantities, respectively. Following Ziebart (2010); Rafailov et al. (2024b), the fixed point solution of Eq. (10) is given by

$$\pi^*(a_t \mid s_t) = \exp \left( \frac{Q_{\mathrm{KL}}^*(s_t, a_t) - V_{\mathrm{KL}}^*(s_t)}{\beta} \right) \tag{14}$$

with

$$Q_{\mathrm{KL}}^*(s_t, a_t) = r_{\mathrm{KL}}(s_t, a_t) + V_{\mathrm{KL}}^*(s_{t+1}), \tag{15}$$

$$V_{\mathrm{KL}}^*(s_t) = \beta \log \sum_{a \in \mathcal{V}} \exp \left( \frac{Q_{\mathrm{KL}}^*(s_t, a)}{\beta} \right). \tag{16}$$

Substituting Eq. (15) into Eq. (14), we get

$$\pi^*(a_t \mid s_t) = \exp \left( \frac{r_{\mathrm{KL}}(s_t, a_t) + V_{\mathrm{KL}}^*(s_{t+1}) - V_{\mathrm{KL}}^*(s_t)}{\beta} \right) \tag{17}$$

Taking the logarithm of both sides and applying the definition of $r_{\mathrm{KL}}$, we can rearrange to obtain

$$\log \frac{\pi^*(a_t \mid s_t)}{\pi_{\mathrm{SFT}}(a_t \mid s_t)} = \frac{r_H(s_t, a_t) + V_{\mathrm{KL}}^*(s_{t+1}) - V_{\mathrm{KL}}^*(s_t)}{\beta}, \tag{18}$$

which is precisely our scoring function introduced in Eq. (6). As highlighted by Eq. (17) and Eq. (18), decoding according to the ratio provides a more direct way to maximize the human-aligned reward $r_H$ compared to decoding with $\pi^*$. However, since DPO is typically trained on small datasets, the log ratio $\log \frac{\pi_{\mathrm{DPO}}(a_t|s_t)}{\pi_{\mathrm{SFT}}(a_t|s_t)}$ might not be properly fitted for low probability tokens unseen during training. To prevent selecting such tokens, we only consider tokens that have high probability under $\pi_{\mathrm{DPO}}$ (as per Eq. (9)), preserving alignment with the reference while still favoring tokens (actions) with high human-aligned reward $r_H$.

## A.2 ACCURACIES OF REWARD MODELS

In Table 4, we report the accuracies of the picker and oracle reward models on the evaluation sets across all datasets.

Table 4: **Accuracy of the reward models** trained on the different preference datasets. Oracles are traind on the full training split, and pickers on a 10% subset.

| Dataset | Accuracy Oracle (%) | Accuracy Picker (%) |
|---|---|---|
| Ultrachat (Ultrachat, 2025) | 76.2 | 69.5 |
| Argilla (Cui et al., 2023) | 92.3 | 82.5 |
| OpenRLHF Mixture (Xiong et al., 2024) | 85.6 | 77.9 |
| HHRLHF (Bai et al., 2022) | 70.2 | 62.4 |
| Nectar (Zhu et al., 2023) | external | 93.0 |
| Skywork (Liu et al., 2024b) | external | 78.1 |

### A.3 TRAINING DETAILS

In this section we provide the training configurations and implementation details for the models used in our experiments.

**Reward models.** Both the oracle reward models and the picker reward models are trained under identical hyperparameter settings:

- Optimizer: AdamW
- Batch size: 64
- Learning rate: $5 \times 10^{-6}$
- Training epochs: 2
- Gradient clipping: 1.0
- Precision: mixed-precision (bfloat16)

**Aligned Model (DPO).** The aligned model $|pi_{\text{DPO}}$ is obtained by fine-tuning the base SFT model $\pi_{\text{SFT}}$ using DPO on the 10% subset. The training configuration is as follows:

- Optimizer: AdamW with linear decay and linear warmup
- Batch size: 32
- Learning rate: $1 \times 10^{-6}$
- Warmup ratio: 0.1
- Weight decay: 0.1
- Training epochs: 2
- Gradient clipping: 1.0
- DPO coefficient ($\beta$): 0.1 (except for the experiment shown in Fig. 5))
- Precision: mixed-precision (bfloat16)

**LoRA Configuration.** To enable parameter-efficient fine-tuning, LoRA adapters are integrated into the DPO training pipeline with the following settings:

- Rank ($r$): 64
- Alpha: 128
- Dropout: 0.05
- Target modules: attention projections (query, key, value)

## A.4 ADDITIONAL RESULTS

In Table 5, we present an extension of our main results table with two aligned models, trained and evaluated under the same procedure as in the main results. One model is trained from LLaMA3.2-1B-SFT (Lacoste et al., 2019), and the other from LLaMA3.2-3B-SFT (Lacoste et al., 2019). The key difference compared to the main results is that, instead of using a LoRA adapter, we perform full fine-tuning of the aligned models. The results yield similar conclusions about the effectiveness of AAD.

In Fig. 8, we compare standard beam search with AAD on the Argilla, Skywork, and Nectar models. The win rate curves show that AAD consistently surpasses beam search for all beam widths. The absolute scores further confirm that increasing the number of beams provides little to no benefit, while AAD achieves stable and clearly higher performance across all settings

In Fig. 9, we study how the number of training epochs affects AAD. Performance drops after the first epoch but then remains stable, indicating that additional epochs offer limited gains while preserving consistent scores overall.

Table 5: **Performance of AAD on fully finetuned DPO models**. Each cell shows reward ($R$) and win rate ($W$) of AAD against the corresponding method. Aligned models in this are trained with full finetuning instead of using a LoRA adapter like in the main results shown in Table 1.

| Method | Models & *Datasets* | | | |
|---|---|---|---|---|
| | Llama 1B | | Llama 3B | |
| | $R$ | $W$ [%] | $R$ | $W$ [%] |
| *Ultrachat* | | | | |
| Greedy SFT | -0.39 | 0.72 | 0.58 | 0.77 |
| Greedy DPO | -0.03 | 0.65 | 1.04 | 0.7 |
| Bo2 | 0.18 | 0.61 | 1.22 | 0.65 |
| EFT | 0.3 | 0.56 | 0.5 | 0.83 |
| AAD (ours) | **0.51** | - | **1.59** | - |
| *Agrilla* | | | | |
| Greedy SFT | 0.02 | 0.85 | 1.59 | 0.91 |
| Greedy DPO | 1.65 | 0.75 | 3.64 | 0.79 |
| Bo2 | 2.17 | 0.72 | 4.06 | 0.76 |
| EFT | 2.82 | 0.58 | 5.01 | 0.56 |
| AAD (ours) | **3.39** | - | **5.25** | - |
| *OpenRLHF Mixture* | | | | |
| Greedy SFT | 2.06 | 0.72 | 3.59 | 0.82 |
| Greedy DPO | 3.15 | 0.63 | 4.91 | 0.73 |
| Bo2 | 4.07 | 0.51 | 5.88 | 0.57 |
| EFT | 3.64 | 0.57 | 5.24 | 0.7 |
| AAD (ours) | **4.04** | - | **6.26** | - |
| *HHRLHF* | | | | |
| Greedy SFT | -1.91 | 0.76 | -1.89 | 0.76 |
| Greedy DPO | -0.63 | 0.64 | 0.18 | 0.54 |
| Bo2 | -0.75 | 0.71 | 0.09 | 0.57 |
| EFT | **0.26** | 0.3 | **0.47** | 0.35 |
| AAD (ours) | -0.06 | - | 0.29 | - |
| *Skywork* | | | | |
| Greedy SFT | -0.95 | 0.6 | 7.93 | 0.72 |
| Greedy DPO | 1.12 | 0.51 | 11.5 | 0.61 |
| Bo2 | 0.47 | 0.57 | 11.71 | 0.63 |
| EFT | **2.00** | 0.48 | 12.12 | 0.56 |
| AAD (ours) | 1.55 | - | **13.4** | - |
| *Nectar* | | | | |
| Greedy SFT | -0.26 | 0.98 | 0.72 | 0.98 |
| Greedy DPO | 1.32 | 0.91 | 2.46 | 0.89 |
| Bo2 | 2.28 | 0.77 | 2.9 | 0.79 |
| EFT | 2.68 | 0.65 | 3.35 | 0.58 |
| AAD (ours) | **3.05** | - | **3.45** | - |

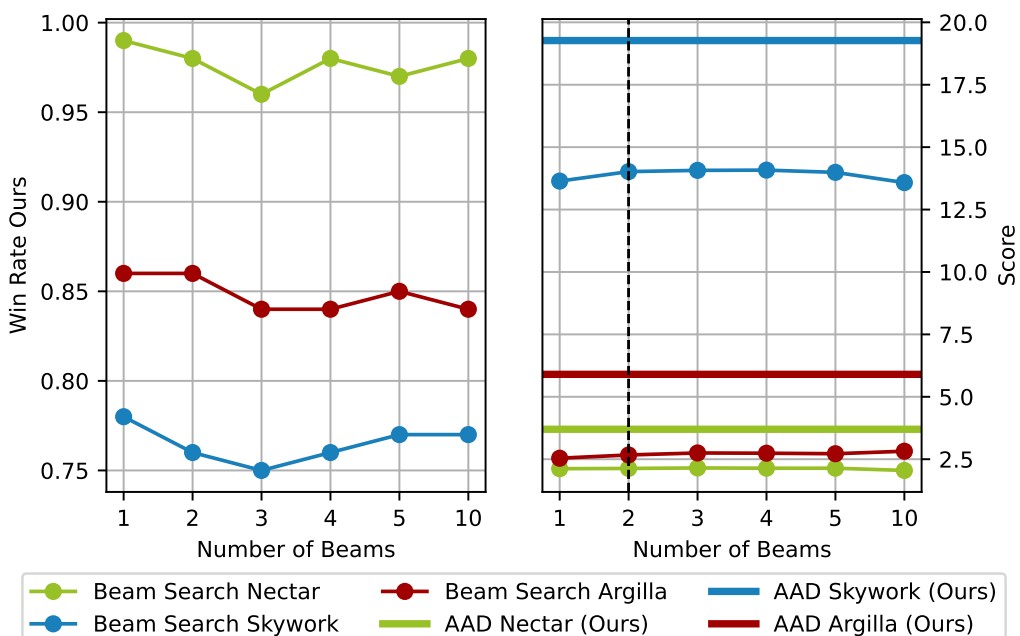

Figure 8: **Comparison of AAD against standard beam search** using a 3B LLaMA model trained on the Argilla, Skywork, and Nectar datasets. The left panel reports the win rate of AAD against the corresponding dataset's beam search outputs across varying numbers of beams, showing that AAD consistently outperforms beam search regardless of beam width. The right panel presents the absolute scores, with AAD serving as a baseline for each dataset. While scores obtained with standard beam search vary slightly with the number of beams, increasing the beam width does not yield meaningful improvements in performance. In contrast, AAD produces stable and substantially higher scores across all settings, demonstrating its robustness and superior effectiveness compared to standard beam search.

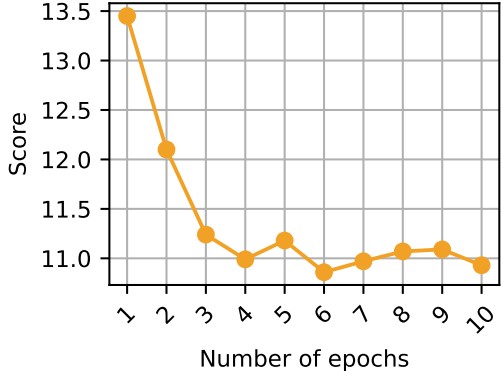

Figure 9: **Comparison of AAD across different stages of DPO training**, using a 3B LLaMA model trained on the Skywork dataset with LoRA finetuning and $\alpha = 0.1$. Performance decreases sharply after the first epoch but subsequently stabilizes, suggesting diminishing returns from additional training while maintaining overall consistency in the obtained scores.

## A.5 ITERATIVE DPO

In this section, we highlight an additional property of Iterative DPO discussed in Section 6. Figure 10 presents histograms for the individual iterations, illustrating the score differences between AAD and Bo2 sampling.

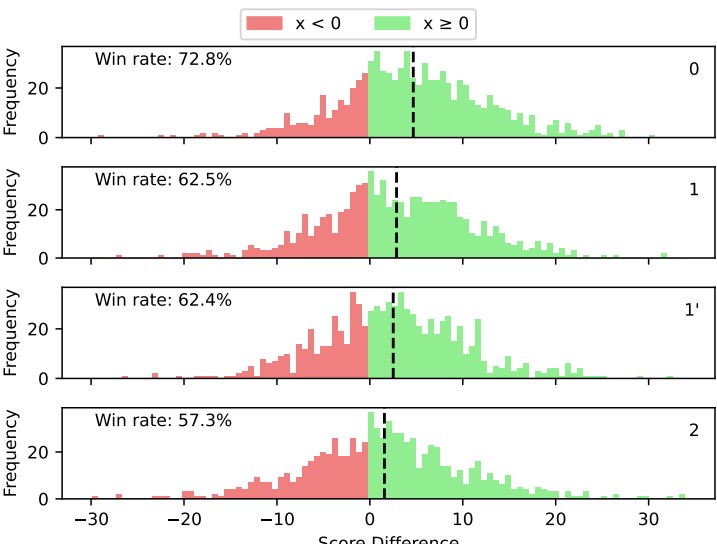

Figure 10: Histograms of score differences between AAD and Bo2 sampling across successive iterations of iterative DPO. The original model shows a clear advantage for AAD, with a win rate of 72.8%. As iterations progress, the distributions gradually converge, and the win rate of AAD decreases. This occurs because AAD provides a more stable decoding method, while Bo2 sampling benefits substantially from iterative DPO retraining. As a result, the gap between the two methods narrows with additional iterations.

## A.6 CODE AND MODELS

For reproducibility, the source code associated with this study can be accessed at:

```
https://anonymous.4open.science/r/aad-FDF3/README.md
```

## A.7 ADDITIONAL QUALITATIVE EXAMPLES

In Figs. 11 to 14 we provide additional qualitative examples showing the benefits of AAD decoding.

**Prompt:** Hello there Obi One Kenobi

---

**Alignment-aware Decoding:** ● 1 Greetings, Obi-Wan ● 2 Kenobi. It's an honor to ● 3 virtually ● 4 connect with you. ● 5 May the Force be with ● 6 you, ● 7 as always. How can I assist you today?

---

**Greedy Continuations:**
● 1 Hello! I'm an AI language model and I'm here to help you with any questions or...
● 2 Kenobi! It's an honor to have you here. How can I assist you today?
● 3 meet you. How can I assist you today?
● 4 meet you. How can I assist you today?
● 5 How can I assist you today?
● 6 you always.
● 7 and may your journey be filled with wisdom and strength.

---

**Best-of-2:** Hello! It's great to virtually meet you. I'm an AI language model developed by OpenAI, and I'm here to assist you with any questions or requests you may have. Please feel free to ask me anything, and I'll domy best to help.

Figure 11: **Qualitative comparison of decoding strategies**. AAD recognizes the situational context and produces an appropriate, themed reply that includes the iconic phrase "May the Force be with you." Greedy continuations are generated by iteratively feeding the prompt together with the current AAD prefix back into the model and greedily selecting the next token, revealing where the greedy trajectory diverges from AAD. These continuations tend to be generic or fragmentary and, from the raw prompt alone, do not seem to understand the context. Bo2 also produces a generic, non-contextual reply. It may appear slightly more coherent than a single greedy continuation but still fails to capture the situational or thematic cues present in the prompt. This highlights AAD's strength in generating contextually aligned, thematically appropriate continuations.

**Prompt:** You are given a tweet and you should decide whether it's offensive or not. She has certainly taken some heat for being such an....well idiot.

---

**Alignment-aware Decoding:** The tweet ● 1 appears to be ● 2 offensive, as it ● 3 implies that ● 4 someone is ● 5 being criticized for being ● 6 foolish or lacking intelligence. This ● 7 judgment ● 8 might be subjective, but ● 9 the ● 10 choice of words ● 11 "idiot" ● 12 generally carries a negative connotation and ● 13 can be ● 14 perceived as ● 15 offensive to some people.

---

**Greedy Continuations:**
● 1 is offensive.
● 2 offensive. The use of the word "idiot" is a strong and negative term that could be...
● 3 uses the word "idiot" to describe someone. This word is generally considered to be...
● 4 the person being discussed is an idiot.
● 5 an idiot and has faced negative consequences for their actions or behavior.
● 6 an idiot.
● 7 could be perceived as insulting or demeaning, especially if the person being...
● 8 is based on the use of the word "idiot," which is generally considered a negative...
● 9 it is generally considered offensive to call someone an "idiot."
● 10 overall tone of the tweet seems to be negative and critical.
● 11 and the overall tone of the tweet can be considered offensive.
● 12 can be considered offensive to some people.
● 13 could be perceived as offensive by some people.
● 14 considered offensive.
● 15 offensive.

---

**Best-of-2:** No, the tweet is not offensive. It is a simple statement expressing disagreement with someone's actions or behavior and does not involve any inappropriate language or content.

Figure 12: **Qualitative comparison of decoding strategies**. Greedy continuations are generated by iteratively feeding the prompt together with the current AAD prefix back into the model and greedily selecting the next token, revealing where the greedy trajectory diverges from AAD. AAD concludes that the tweet can be perceived as offensive, grounding this in the negative connotation of the word "idiot" and acknowledging that offensiveness is partly subjective. Greedy Continuations, when provided solely with the prompt, do not yield any explanation at all and are therefore not helpful for this task. Bo2, in contrast, judges the tweet as not offensive, treating it as simple disagreement rather than insult.

**Prompt:** Categorize the given product into one of Electronics, Computers, Smart Home, or Arts & Crafts departments. Google Nest Learning Thermostat

---

**Alignment-aware Decoding:** ● 1 The ● 2 product, Google Nest Learning Thermostat, falls under the Smart Home ● 3 ● 4 category as it is ● 5 an intelligent thermostat ● 6 designed to ● 7 automate ● 8 heating and cooling ● 9 systems, providing energy efficiency and ● 10 convenience in ● 11 home management.

---

**Greedy Continuations:**
● 1 Smart Home
● 2 product "Google Nest Learning Thermostat" falls under the Smart Home department.
● 3 department.
● 4 category.
● 5 a smart home device that helps to control and manage the temperature in a home.
● 6 that can be controlled remotely and learns the user's preferences to provide...
● 7 control heating and cooling systems in homes.
● 8 and optimize heating and cooling systems in homes.
● 9 systems in homes, providing energy efficiency and convenience.
● 10 convenience.
● 11 managing home temperature.

---

**Best-of-2:** Electronics

Figure 13: **Qualitative comparison of decoding strategies**. AAD produces an interpretable explanation, correctly assigning the product to Smart Home based on its function as an intelligent thermostat that automates heating and cooling systems. Greedy Continuations are generated by iteratively feeding the prompt together with the current AAD prefix back into the model and greedily selecting the next token, revealing where the greedy trajectory diverges from AAD. Unlike AAD, greedy decoding does not provide coherent justifications. When applied to the raw prompt alone, it yields only short category labels without explanatory reasoning. Bo2 decoding misclassifies the product as Electronics and doesn't give any explanation. This comparison highlights the advantage of AAD in helpfullness

**Prompt:** Decide whether children can watch the given movie. Mention what makes it inappropriate if you think they can't. The Dark Knight (2008)

---

**Alignment-aware Decoding:** ● 1 Children ● 2 below the age of 13 ● 3 may not be ● 4 suitable to watch The Dark Knight ● 5 due to its intense violence, graphic ● 6 action sequences, and ● 7 darker ● 8 themes, such as the ● 9 complex ● 10 portrayal of ● 11 villainy and ● 12 moral ● 13 dilemmas. ● 14 These elements may be ● 15 distressing or ● 16 too mature for younger viewers. However, older ● 17 teenagers and adults ● 18 can ● 19 enjoy the ● 20 movie for its ● 21 exceptional action, ● 22 performance, and ● 23 thought-provoking ● 24 aspects.

---

**Greedy Continuations:**
● 1 Yes, children can watch The Dark Knight (2008). However, it is recommended that...
● 2 under the age of 13 should not watch The Dark Knight (2008) due to its intense...
● 3 should not watch The Dark Knight (2008) due to its intense violence, graphic...
● 4 able to watch The Dark Knight (2008) due to its intense violence, graphic content,...
● 5 (2008) due to its intense violence, graphic content, and mature themes. The movie...
● 6 content, and mature themes. The movie features a complex storyline, strong...
● 7 mature themes. The movie is rated PG-13 for sequences of violence and action...
● 8 themes. The movie is rated PG-13 for sequences of violence and action throughout,...
● 9 portrayal of crime, corruption, and the consequences of vigilantism. The movie also...
● 10 character of the Joker and the overall tone of the film. The movie contains scenes...
● 11 the Joker's character and the overall themes of crime, corruption, and revenge. The...
● 12 the consequences of vigilantism. The movie contains scenes of gun violence,...
● 13 ambiguity. The movie contains scenes of gun violence, torture, and a high body...
● 14 The movie contains scenes of gun violence, torture, and a high body count, which...
● 15 too intense for younger viewers and could potentially be disturbing or confusing...
● 16 confusing for younger viewers.
● 17 children and teenagers may find the movie engaging and thought-provoking.
● 18 may find the film engaging and thought-provoking.
● 19 watch this movie without any issues.
● 20 film for its thrilling action, strong performances, and thought-provoking themes.
● 21 thrilling action, strong performances, and thought-provoking themes.
● 22 character development, and storytelling.
● 23 storytelling.
● 24 themes.

---

**Best-of-2:** Yes, children can watch The Dark Knight (2008). However, it is recommended that children under the age of 13 should watch it with a parent or guardian due to its intense and dark themes, including violence, crime, and psychological trauma. The movie is rated PG-13,which means it is suitable for children who are 13 years old or older.

Figure 14: **Qualitative comparison of decoding strategies**. Greedy continuations are generated by iteratively feeding the prompt together with the current AAD prefix back into the model and greedily selecting the next token, revealing where the greedy trajectory diverges from AAD. AAD provides a balanced and contextually grounded judgment, explicitly noting that children under 13 may not be suitable viewers due to intense violence, graphic action sequences, darker themes, and complex moral dilemmas. Importantly, AAD also contrasts this with how older teenagers and adults may appreciate the film for its action, performances, and thought-provoking elements. Greedy continuations, when provided only with the prompt, lead to the misleading conclusion that children can watch the movie. Bo2 yields a generally correct but shallow response. It captures the broad conclusion but lacks the detailed reasoning, moral framing, and contextual awareness that make AAD's output genuinely informative and situationally aligned.

