# OpenReview forum: "Alignment-Aware Decoding"
_ICLR.cc/2026/Conference — Submitted to ICLR 2026_

### Official Review · Reviewer_1wWf · 2025-10-24

**Soundness:** 3
**Presentation:** 3
**Contribution:** 2
**Rating:** 4
**Confidence:** 3

**Summary:**

The paper addresses current issues with common alignment techniques such as RLHF and DPO by proposing a new method for implicit reward optimization.

**Strengths:**

- Strong experimental section with multiple LLMs and a diverse set of baselines.
- The method consistently outperforms prior work across several datasets.
- Good ablation studies that help isolate the effects of different components in the pipeline.
- Empirical results are reproducible and sufficiently robust.

**Weaknesses:**

- The method section is a bit unclear, and more importantly how the modification can fit into the overall algorithm.
- No theoretical analysis is provided to support claims about implicit optimization or stability.
- The use of *Best-of-2* as a decoding strategy is limiting; larger \(N\) values would strengthen the conclusions, especially on the Nectar datasets, similar to what was done for Argilla and Skwy
- Missing comparisons with alternative controllable generation strategies such as controlled decoding [1] and strong search-based methods (e.g., beam search). We understand that beam search cannot be used in conjunction with this method, but maybe as a baseline.

---

**References**
[1] Mudgal, Sidharth, et al. "Controlled decoding from language models." arXiv preprint arXiv:2310.17022 (2023).

**Questions:**

**Questions for the Authors**
1. You state that Best-of-\(N\) will eventually outperform for larger \(N\), but suffers from increased compute. Can you quantify this trade-off?
2. Can you explain more clearly how Equations (7)–(9) integrate into the overall optimization framework? It is not obvious how they operationalize the main objective

**Suggestions for Improvement**
- Add clear algorithmic pseudocode.
- Spend more effort explaining intuition and the core method instead of re-describing RLHF and DPO (which readers already know).
- Improve Figure 1 explanation. The color codes are not very clear.
- Add theoretical discussion or at least a more formal insight into why the method should work.

---

> ### Author Response · Authors · 2025-11-25
>
> We thank the reviewer for their valuable feedback and address each concern below.
>
> > Can you explain more clearly how Equations (7)–(9) integrate into the overall optimization framework? It is not obvious how they operationalize the main objective
>
> AAD can be seen as an inference time optimization. Eq. (7) computes the token-level rewards that the log ratio of the DPO and SFT models approximates. Selecting the highest reward at each step is a form of greedy reward optimization. However, due to numerical instabilities or model under/overfitting, tokens with low probabilties might be subject to large ratios. To avoid this, Eq. (8) and (9) disregard tokens with small probabilities under $\pi_{dpo}$ using min-$\alpha$ filtering.
>
> > No theoretical analysis is provided to support claims about implicit optimization or stability.
>
> > Add theoretical discussion or at least a more formal insight into why the method should work.
>
> We have added a detailed theoretical discussion about AAD in Appendix A.1. The original formulation of DPO casts it as a bandit problem, that is, DPO trains a sequence-level reward model parametrized by $\pi_{dpo}(y|x)/\pi_{sft}(y|x)$. However, at inference, this model is decoded autoregressively, introducing a train-test mismatch. As mentioned in section 4, this DPO model also inherits the biases of the SFT version due to the KL constraint.  In other words, DPO does not train an optimal policy that can be sampled to achieve maximal reward. On the other hand, [1] show that DPO trained models are capabale of performaing credit assignment in order to quantify which token contribute more to the overall quality of a text. This is precisely done by computing $\pi_{dpo}(y_t|x,y_{<t})/\pi_{sft}(y_t|x,y_{<t})$.
>
> > The use of _Best-of-2_ as a decoding strategy is limiting; larger (N) values would strengthen the conclusions, especially on the Nectar datasets, similar to what was done for Argilla and Skwy
>
> We have added the BoN study for Nectar in Figure 2. Results show that AAD beats BoN up to $N=50$, even when using the oracle (i.e., the evaluator) as the picker model for BoN. In Table 1, we only report Bo2 in order to allow a fair (i.e., compute-equivalent) comparison with AAD.
>
> > Missing comparisons with alternative controllable generation strategies such as controlled decoding and strong search-based methods (e.g., beam search). We understand that beam search cannot be used in conjunction with this method, but maybe as a baseline.
>
> We have added comparisons with contrastive decoding (see Table 3) and with beam search (see Figure 8). AAD clearly outperforms the two new baselines, providing evidence of its effectiveness. We did not compare controllable decoding, as the authors’ code is not publicly available. However, we note that EFT, which we extensively compare to, can be seen as a variant of controllable decoding, where $V_\theta = \frac{\pi_{dpo}}{\pi_{sft}}$.
>
> > You state that Best-of-(N) will eventually outperform for larger (N), but suffers from increased compute. Can you quantify this trade-off?
>
> The compute of Best-of-N scales linearly with $N$. Let $\epsilon$ be the total probability of sampling sequences that exceed the score of the completion generated by AAD for a given prompt, using the DPO-aligned model. The expected number of samples to get one such sequence is $\mathbb{E}[N] = 1/\epsilon$. Hence, the expected compute ratio between BoN and AAD is $1/2\epsilon$.
>
> > Spend more effort explaining intuition and the core method instead of re-describing RLHF and DPO (which readers already know).
>
> > The method section is a bit unclear, and more importantly how the modification can fit into the overall algorithm.
> >
> > Add clear algorithmic pseudocode.
>
> We thank the reviewer for their feedback concerning the clarity of our work. We have added an algorithmic pseudocode and have revised the method section.
>
> > Improve Figure 1 explanation. The color codes are not very clear.
>
> In Figure 1, we qualitatively compare AAD with both greedy and Bo2 on the DPO model.  For instance, at position 8, given a context up to `[...] can weigh`, AAD generates `between 2.5 to 6 pounds [...]` while greedy generates `anywhere from 2 to 6 pounds [...]`. The color code is simply there to easily distinguish the completions. We have updated Figure 1's caption to improve clarity.
>
> We thank the reviewer for the suggested improvemenent and we hope these additions will increase the reviewer's confidence in the overall quality of our work.
>
> [1] Rafael Rafailov et al., From $r$ to $Q^*$: Your language model is secretly a Q-function. COLM, 2024.

---

### Official Review · Reviewer_XYjx · 2025-10-29

**Soundness:** 2
**Presentation:** 2
**Contribution:** 2
**Rating:** 4
**Confidence:** 4

**Summary:**

This paper introduces Alignment-Aware Decoding, a simple and training-free inference method to improve the alignment of large language models. AAD operates by using the token-level log-likelihood ratio between a DPO-aligned model and its SFT reference model as an implicit reward signal to guide text generation. The core contributions are its simplicity and strong empirical results, demonstrating consistent outperformance against strong baselines like Best-of-N sampling across multiple benchmarks and model scales. This paper shows AAD can generate high-quality synthetic data to effectively enhance model alignment through iterative DPO, providing a practical solution for data-scarce scenarios.

**Strengths:**

1. The paper's core strength is its simplicity. It proposes a clean, training-free method that cleverly leverages existing models from the DPO training process to enhance alignment.
2. The experimental evidence is consistent. AAD consistently outperforms standard decoding baselines across multiple model families and benchmarks.
3. A major contribution is its application in synthetic data generation. The paper demonstrates that data generated by AAD can nearly close the performance gap in low-resource scenarios, which is a highly practical and valuable finding.

**Weaknesses:**

1. The paper's primary weakness is its limited novelty. The core mechanism of AAD is using the log-probability ratio between an 'expert' model and an 'amateur' reference model to guide generation, which is conceptually almost identical to Contrastive Decoding (VCD) and its variants.
2. The paper fails to include any direct experimental comparison against VCD or its variants. Without this crucial baseline, it is impossible to assess whether AAD offers any tangible benefits over existing techniques.
3. The method requires two forward passes, doubling the inference cost, which is prohibitive for latency-sensitive applications.

**Questions:**

1. Is the specific SFT model strictly necessary as the 'amateur' model, or could it be replaced with a generic, weaker model or a noise-corrupted model, similar to VCD? This is crucial for understanding AAD's novelty.
2. Have you compared AAD's performance against a standard VCD setup using a strong model against a weak model? This is essential to isolate whether the gain comes from DPO's alignment signal or just the contrast between any strong and weak model.
3. The 2x inference cost is a major limitation. Have you explored any mitigation strategies that don't require two forward passes at runtime?

---

> ### Author Response · Authors · 2025-11-25
>
> We thank the reviewer for their valuable feedback and address each concern in the following.
>
> > 1. The paper's primary weakness is its limited novelty. The core mechanism of AAD is using the log-probability ratio between an 'expert' model and an 'amateur' reference model to guide generation, which is conceptually almost identical to Contrastive Decoding (VCD) and its variants.
>
> > 2. The paper fails to include any direct experimental comparison against VCD or its variants. Without this crucial baseline, it is impossible to assess whether AAD offers any tangible benefits over existing techniques.
>
> > 1. Is the specific SFT model strictly necessary as the 'amateur' model, or could it be replaced with a generic, weaker model or a noise-corrupted model, similar to VCD? This is crucial for understanding AAD's novelty.
>
> > 2. Have you compared AAD's performance against a standard VCD setup using a strong model against a weak model? This is essential to isolate whether the gain comes from DPO's alignment signal or just the contrast between any strong and weak model.
>
> We understand that the reviewer's main concern is the lack of comparison with contrastive decoding. To address this, we have added a direct comparison between contrastive decoding (CD) and AAD in order quantify the gains obtained by AAD (see below or Table 3 of the revised manuscript).
>
> | Method | Argilla R | Argilla W | Skywork R | Skywork W | Nectar R | Nectar W |
> |-|:-:|:-:|:-:|:-:|:-:|:-:|
> | CD with Llama 1B   | 2.79 | 0.85 | 13.08 | 0.81 | 2.65 | 0.9 |
> | CD with Llama 3B   | 2.46 | 0.86 | 10.88 | 0.88 | 2.51 | 0.93 |
> | AAD    | **5.9** | - | **19.27** | - | **3.7** | - |
>
> As shown by the significant increase in average reward, and in accordance with the theory, the SFT model (Llama 8B) _cannot_ be replaced with a different amateur model (Llama 1B or 3B) without a significant loss in alignment. Furthermore, the high win-rate (consistantly $>80$%) provides strong evidence that AAD offers tangible benefits over existing techniques. We believe that these substantial gains over CD (win-rates consistantly above 80%), together with the iterative DPO experiments (eliminating the need for two forward passes), the theoretical analysis (linking to the literature, see Appendix A.1), and the extensive empirical evaluation underscore the technical novelty and significance of our work.
>
> > 3. The method requires two forward passes, doubling the inference cost, which is prohibitive for latency-sensitive applications.
>
> Since the two forward passes are independent, they can either run in parallel, which keeps latency unchanged but requires twice the hardware, or sequentially, which doubles latency but needs no extra hardware. AAD is therefore _not_ prohibitive in latency-sensitive applications, and remains flexible depending on deployment constraints. The sampling stage, which includes the log ratio, min-$\alpha$ filtering, and token selection, adds only a negligible cost that is comparable to that of most other sampling stategies.
>
> > 3. The 2x inference cost is a major limitation. Have you explored any mitigation strategies that don't require two forward passes at runtime?
>
> We agree that the 2x inference cost can be a limitation, although it is a weakness that is shared by other decoding strategies such as CD. We have indeed explored such mitigation strategies. In section 6, we use iterative DPO to generate a synthetic dataset with AAD and train a new DPO model on this synthetic data. Results are shown in Figure 6. Remarkably, after two iterations of this procedure, we see that the obtained model achieves a 68.2% win-rate against the original DPO model when both are using greedy decoding. Even more remarkably, this method outperfoms the DPO model aligned on 100% on the data, but using only 10% of the data (68.2% win-rate against baseline vs 63.85%, using greedy decoding).
>
> We hope the newly added experiments, particularly the comparison with contrastive decoding, reinforce the reviewer’s confidence in the efficacy of our method. We remain happy to provide any further details or analyses if needed.

---

### Official Review · Reviewer_gJtS · 2025-11-01

**Soundness:** 3
**Presentation:** 3
**Contribution:** 3
**Rating:** 6
**Confidence:** 4

**Summary:**

This paper introduces Alignment-Aware Decoding (AAD), a method to enhance the alignment of models already trained with Direct Preference Optimization (DPO). AAD operates at inference time by using the token-level probability ratio between the DPO model and its Supervised Fine-Tuning (SFT) base model as an implicit reward to guide the decoding process. Experimental results demonstrate that AAD significantly outperforms baselines such as greedy decoding, best-of-N (BoN), and Emulated Fine-Tuning (EFT) across several alignment benchmarks. The paper also shows that synthetic data generated by AAD can be used for iterative DPO training, creating a positive feedback loop to further improve alignment.

**Strengths:**

AAD is an inference-time technique. Its computational overhead is approximately twice that of greedy decoding, yet it demonstrates performance that surpasses other methods with similar computational costs.

The method proves effective even when the DPO model is trained on only 10% of the available preference data, highlighting its utility in low-data regimes. Moreover, synthetic samples generated by AAD can be fed back into the DPO process, to further boost model alignment.

**Weaknesses:**

1, What is the effect of plausible token ranges determined by min-$\alpha$ filtering? Ablation studies of $\alpha$ are needed.

2, The main results (e.g., in Table 1) appear to be based on DPO models trained with only 10% of the training data . It is unclear whether AAD provides a similar performance boost over Greedy DPO and other baselines when all available training data is used for the DPO training.

3, While being backed by reasons, the core idea of contrasting the DPO model with the SFT model at a token level bears a strong resemblance to contrastive decoding . This similarity may diminish the perceived technical novelty of the work.

**Questions:**

1. The method relies on a "properly fitted" DPO model. In practice, how can one determine the optimal point to stop DPO training before applying AAD or training on AAD-generated data? Furthermore, what is the relationship between the quality of the DPO model (e.g., trained on more data or for more steps) and the relative gains from AAD? Does AAD provide diminishing returns as the base DPO model becomes stronger?
2. In Figure 2, the performance gap between BoN-Oracle and BoN-Picker widens much more significantly on the Skywork dataset than on the Argilla dataset as N increases. Could the authors provide more dataset-specific analysis or insights to explain this?
3. Presentation issue: The last sentence on Page2 appears to be unfinished.

---

> ### Author Response · Authors · 2025-11-25
>
> We thank the reviewer for their valuable feedback and address each concern below.
>
> >1, What is the effect of plausible token ranges determined by min-$\alpha$ filtering? Ablation studies of $\alpha$ are needed.
>
> We have added an ablation study for the $\alpha$ parameter (see Figure 3 in the revised manuscript). We observe that AAD obtains the best performance at around 0.1, which we use for all our experiements. Smaller values allow too many tokens to be considered for the alignment optimization, while larger values overly constrain it. For $\alpha=1$, one recovers standard greedy decoding.
>
> > 2, The main results (e.g., in Table 1) appear to be based on DPO models trained with only 10% of the training data . It is unclear whether AAD provides a similar performance boost over Greedy DPO and other baselines when all available training data is used for the DPO training.
>
> We would like to direct the reviewer to Figure 4, which already includes an ablation on training dataset size, up to 100%. As shown, win-rates remain largely stable despite modest score increases, indicating that AAD provides consistent performance gains across dataset scales. For our main results, we evaluate AAD on 10% of the data to ensure that the _evaluator_ (oracle), which has seen 100%, remains stronger than the _evaluee_ (AAD and baselines).
>
> > 3, While being backed by reasons, the core idea of contrasting the DPO model with the SFT model at a token level bears a strong resemblance to contrastive decoding . This similarity may diminish the perceived technical novelty of the work.
>
> We have added a direct comparison between contrastive decoding (CD) and AAD in order quantify the gains obtained by AAD (see Table 3 of the revised manuscript). We believe that the substantial gains over CD (win-rates consistantly above 80%), together with the iterative DPO experiments (eliminating the need for two forward passes), the theoretical analysis (see Appendix A.1), and the extensive empirical evaluation, underscore the technical novelty and significance of our work.
>
> > 1. The method relies on a "properly fitted" DPO model. In practice, how can one determine the optimal point to stop DPO training before applying AAD or training on AAD-generated data? Furthermore, what is the relationship between the quality of the DPO model (e.g., trained on more data or for more steps) and the relative gains from AAD? Does AAD provide diminishing returns as the base DPO model becomes stronger?
>
> We thank the reviewer for these insightful questions. Regarding the data used to train the DPO model, we refer to our response above and to Figure 4. As for training steps, AAD theoretically performs best when properly fitted and is inherently robust to overfitting, since the token-level reward ordering remains largely unaffected even if the DPO model overfits. This behavior is confirmed by our new experiments in Appendix A.4 (Figure 9), which shows that after an initial drop between epoch 1 and 3, AAD scores stabilize.
>
> > 2. In Figure 2, the performance gap between BoN-Oracle and BoN-Picker widens much more significantly on the Skywork dataset than on the Argilla dataset as N increases. Could the authors provide more dataset-specific analysis or insights to explain this?
>
> We believe there is a simple explanation. The skywork reward oracle is an external pretrained checkpoints that ranked first on RewardBench. It has been trained using a series of tricks for reward modeling [1], and might therefore be misaligned with the data used to train corresponding picker and DPO models, which we have trained on 10% of the original data with our own training schemes. While the picker still achieves relatively high accuracy (78.1% as shown in Table 4), it may deviate from the oracle reward due to different training pipelines. In contrast, for Argilla, we have trained both model (oracle and picker), changing only the dataset size (100% and 10%, respectively). We emphasize the the comparisons with BoN with oracles are biased toward BoN, since in that setting, the oracale is used both as the picker and as the evaluator. In other words, BoN has access to the oracle while AAD doesnt. In addition, we have added the results for the Nectar dataset, showing a similar trend than Argilla.
>
> > 3. Presentation issue: The last sentence on Page2 appears to be unfinished.
>
> Apologies, we have fixed the sentence in the revised manuscript.
>
> We hope these clarifications and added experiements will further reinforce the reviewer’s positive assessment and are happy to respond to any further inquiries.
>
> [1] Liu et al., _Skywork-reward: Bag of tricks for reward modeling in LLMs_, 2024.

---

### Official Review · Reviewer_dygm · 2025-11-01

**Soundness:** 3
**Presentation:** 3
**Contribution:** 3
**Rating:** 8
**Confidence:** 4

**Summary:**

This paper proposes Alignment-Aware Decoding (AAD), a novel inference-time method designed to improve the alignment of large language models without additional training. The core idea is to exploit the log-likelihood ratio between a DPO-trained model and its reference SFT model  as a token-level implicit reward, guiding decoding decisions toward tokens that DPO training implicitly upweighted. By decoding only within a constrained “feasible token subset,” AAD mitigates instability and mode collapse while effectively amplifying alignment signals during inference. The method is theoretically motivated by analyzing the analytical optimum of DPO and empirically validated on multiple preference datasets (Ultrachat, Argilla, HHRLHF, Nectar, Skywork). Results show consistent gains in win rate and average reward over greedy, Best-of-N, and emulated fine-tuning baselines, particularly in low-data or high-β (strong KL constraint) regimes.

**Strengths:**

1.The paper convincingly identifies reference bias as a fundamental weakness of DPO training and derives AAD directly from the analytical form of the DPO optimum. This theoretical clarity distinguishes it from prior heuristic inference-time alignment methods.

2.AAD requires no retraining, no reward model, and minimal computation overhead—only two forward passes (DPO and SFT). The implementation is straightforward yet conceptually powerful, making it easy to adopt in existing alignment pipelines.

3.The empirical study is broad, covering diverse datasets and model scales. AAD consistently improves both reward and human-aligned metrics, outperforming stronger baselines under equal compute.

**Weaknesses:**

1.AAD requires both a DPO-aligned model and its original SFT counterpart. This dependency limits applicability to settings where both checkpoints are available, which might not hold for commercial or closed-source models.

2.Each decoding step requires two model forward passes, doubling inference time. While computationally modest compared to full fine-tuning, a quantitative analysis of runtime overhead would improve clarity.

3.Most results rely on internal or proxy reward models as alignment judges. Additional human evaluation or LLM-as-judge should also be included.

4.The paper could include more diagnostic studies, e.g., where AAD overemphasizes high-ratio tokens that hurt coherence or factual accuracy, to better characterize its limits.

**Questions:**

1.Could AAD be extended to other alignment objectives (e.g., SimPO, ORPO, PPO-trained models), or is it specific to the DPO formulation?

2.Could AAD interact beneficially with diversity-promoting sampling (e.g., nucleus or contrastive sampling), or would that dilute the alignment effect?

---

> ### Author Response · Authors · 2025-11-25
>
> We thank the reviewer for their valuable feedback and address each concern in the following.
>
> > 1.AAD requires both a DPO-aligned model and its original SFT counterpart. This dependency limits applicability to settings where both checkpoints are available, which might not hold for commercial or closed-source models.
>
> We agree that requiring both the DPO-aligned model and its original SFT counterpart may limit AAD’s applicability in some cases, as discussed in the conclusion section of the manuscript. However, even for commercial or closed-source applications, AAD can still be easily implemented by the provider if the SFT model checkpoint is accessible.
>
> > 2.Each decoding step requires two model forward passes, doubling inference time. While computationally modest compared to full fine-tuning, a quantitative analysis of runtime overhead would improve clarity.
>
> Since the two forward passes are independent, they can either run in parallel, which keeps latency unchanged but requires twice the hardware, or sequentially, which doubles latency but needs no extra hardware. This makes the method flexible depending on deployment constraints. The sampling stage, which includes the log ratio, min-$\alpha$ filtering, and token selection, adds only a negligible cost that is comparable to that of most other sampling stategies.
>
> > 3.Most results rely on internal or proxy reward models as alignment judges. Additional human evaluation or LLM-as-judge should also be included.
>
> We agree that external evaluation is important. However, AAD can only optimize toward the preference signal present in the training data. Our experiments focus on assessing how well AAD aligns to that signal. Dataset-agnostic evaluation methods (such as human raters or LLM-as-judge) introduce ambiguity, since poor scores could reflect shortcomings of the underlying dataset rather than limitations of AAD itself. That said, we include an LLM-as-judge evaluation in Table 2 (AlpacaEval with GPT-4 as the judge). Results show that AAD consistantly matches or outperforms baselines.
>
> > 4.The paper could include more diagnostic studies, e.g., where AAD overemphasizes high-ratio tokens that hurt coherence or factual accuracy, to better characterize its limits.
>
> We thank the reviewer for this suggestion. We have investigated such cases but have not yet identified any notable instances. The min-$\alpha$ filtering step is specifically designed to exclude tokens that could harm coherence or factual accuracy. Tokens that pass this threshold could, in principle, also be sampled by standard methods such as top-$k$ or top-$p$, which may explain why such critical cases are rare for AAD.
>
> > 1.Could AAD be extended to other alignment objectives (e.g., SimPO, ORPO, PPO-trained models), or is it specific to the DPO formulation?
>
> This is an interesting question. We are currently running these experiments and will add them to the manuscript once completed. However, we suspect that AAD will not be as effective since these objectives do not directly target the implicit reward expressed as the ratio of the policy and its reference.
>
> > 2.Could AAD interact beneficially with diversity-promoting sampling (e.g., nucleus or contrastive sampling), or would that dilute the alignment effect?
>
> Since AAD is deterministic, which is both a strength and a limitation, its interaction with other sampling strategies is not straightforward. From a theoretical perspective however, modifying the decoding step could indeed dilute the alignment effect. This is an interesting avenue for future work.
>
>
> We hope the clarifications will further strengthen the reviewer’s positive assessment.

---

### Meta-Review · Area_Chair_SDdM · 2026-01-06

**Summary:**

While the paper presents a clean and effective inference-time method with strong empirical performance, multiple reviewers raised persistent concerns about technical novelty relative to prior decoding and test-time scaling work. In particular, reviewers gJtS and XYjx noted that the core mechanism—guiding generation using a token-level log-probability ratio between a stronger and weaker model—bears a close conceptual resemblance to established contrastive decoding and expert–amateur steering approaches, calling into question whether AAD introduces a fundamentally new idea or a DPO-specific instantiation of known techniques. Although the authors addressed this by adding direct comparisons to contrastive decoding and arguing that the SFT reference plays a uniquely principled role in the DPO setting, reviewers remained divided on whether this distinction constitutes a conceptual advance or an incremental specialization. Additional novelty concerns centered on the reliance on familiar decoding trade-offs (two forward passes, feasible-token filtering to stabilize ratios), which reviewers viewed as practical engineering choices rather than new alignment insights. Taken together, the discussion clarifies that AAD is a well-motivated and empirically strong method, but leaves unresolved whether it meaningfully expands the space of inference-time alignment techniques beyond existing decoding-based approaches, which tempers enthusiasm for acceptance on novelty grounds.

**Reviewer Concerns:**

See above.

**Reviewer Scores:**

See above.

---

### Decision · Program_Chairs · 2026-01-26

Reject